# Rainbow Trout (*Oncorhynchus mykiss*) as Source of Multifunctional Peptides with Antioxidant, ACE and DPP-IV Inhibitory Activities

**DOI:** 10.3390/nu15040829

**Published:** 2023-02-06

**Authors:** Martina Bartolomei, Janna Cropotova, Carlotta Bollati, Kristine Kvangarsnes, Lorenza d’Adduzio, Jianqiang Li, Giovanna Boschin, Carmen Lammi

**Affiliations:** 1Department of Pharmaceutical Sciences, Università degli Studi di Milano, Via Luigi Mangiagalli, 25, 20133 Milano, Italy; 2Department of Biological Sciences Ålesund, Norwegian University of Science and Technology, 6009 Ålesund, Norway

**Keywords:** rainbow trout, hydrolysates, bioactive peptides, bioactivities, antioxidant activity, angiotensin converting enzyme inhibitory properties, dipeptidyl peptidase-IV inhibitory properties

## Abstract

The present study aimed at characterizing the possible biological activities of the multifunctional low molecular weight fractions (<3 kDa) peptides isolated from rainbow trout (*Oncorhynchus mykiss*) obtained by enzymatic hydrolysis. The fish protein hydrolysate (FPH) was tested for its antioxidant property along with its angiotensin converting enzyme (ACE) and dipeptidyl peptidase IV (DPP-IV) inhibitory activities. In particular, the 2,2-diphenyl-1-picrylhydrazyl (DPPH), the ferric reducing antioxidant power (FRAP), the oxygen radical absorbance capacity (ORAC) assay and the 2,2′-Azino-bis (3-ethylbenzothiazoline-6-sulfonic acid) diammonium salt (ABTS) assays were carried out for the evaluation of the in vitro antioxidant activity. The cell-free ACE and DPP-IV inhibitory activity assays were also estimated, showing a dose-dependent inhibition. These biological properties were additionally quantified at the cellular level using human intestinal Caco-2 cells. Namely, the antioxidant activity was determined by evaluating the capability of the hydrolysate to reduce the H_2_O_2_-induced reactive oxygen species (ROS) and lipid peroxidation levels, and the DPP-IV activity assays show a reduction of enzyme activity of up to 27.57 ± 3.7% at 5 mg/mL. The results indicate that *Oncorhynchus mykiss*-derived peptides may have potential employment as health-promoting ingredients.

## 1. Introduction

Bioactive peptides are defined as a group of peptides that have a biological function in a cell or a living organism, and they play important roles in physiological functions and pathogenesis. Potentially bioactive peptides can be extracted or produced from any organism [1]. Animal-derived proteins are a significant source of bioactive peptides which are mainly produced by the enzymatic hydrolysis of proteins. An important source of animal protein is marine organisms. In fact, in literature, there is abundant scientific evidence that reports the biological activities of peptides and proteins obtained from marine origin comprising mollusks, crustaceans, and fishery raw material or by-products (head, intestines, skin, and fins) [2]. Nowadays, the byproducts obtained from fish or seafood processing is mainly used as fertilizer or as a component of feed for aquariums and poultry [3,4]. Nevertheless, the marine-derived byproducts provide many bioactive molecules, such as collagen, antioxidants, chitin, and polyunsaturated fatty acids, and are notably a great source of proteins [5,6,7]. The fraction containing proteins is well digestible, and it has been employed to produce various hydrolysates, which present a good amino acid balance, great nutritional value, and good functional properties. Furthermore, by analyzing the purified peptide fractions, it is possible both to characterize the peptides with biological activity and to find the fractions with the highest concentration of bioactive peptides. Hydrolysates derived from marine byproducts are composed primarily of di- and tripeptides, which are absorbed more rapidly than free-form amino acids and much more rapidly than intact proteins [8]. It is well known that marine bioactive peptides show many biological functions, including the inhibition of angiotensin converting enzyme (ACE), antioxidant, immune-modulatory, antimicrobial and antifungal, antidiabetic and anticoagulant activities [8,9,10,11].

In a previous work, we characterized the physicochemical features of peptides isolated from rainbow trout (*Oncorhynchus mykiss*). Some rainbow trout proteins and byproducts have already been shown to produce hydrolysate with antioxidant [12,13,14,15], anticancer [16] and antibacterial activities [17]. In literature, different enzymes, i.e., Alcalase [18,19] alkaline protease, papain [20], neutral protease, Flavourzyme, or trypsin [21], and conditions have been evaluated to produce protein hydrolysate from rainbow trout with different degrees of hydrolysis, characteristics and peptide generation. The fish protein hydrolysate (FPH) was obtained from minced rainbow trout raw material treated with two commercial enzymes, namely papain and bromelain (0.05% *w*/*w* each) [22]. Given the excellent physicochemical characteristics and amino acid composition, further investigations have been carried out to valorize the <3 kDa fraction of FPH. Indeed, to foster the biological activity of the FPH hydrolysate, the first purpose of the present study was an in-depth assessment of its potential antioxidant, anti-DPP-IV, and anti-ACE activities in vitro, respectively. To achieve these objectives, the direct antioxidant activity was tested using different in vitro antioxidant assays, namely, DPPH, ferric reducing antioxidant power (FRAP), and 2,2-azino-bis-(3-ethylbenzothiazoline-6-sulfonic) acid (ABTS), and at the cellular level. In addition, the FPH capability to decrease the activity of both DPP-IV and ACE enzymes was biochemically evaluated. Then, the FPH antioxidant activity was measured by assessing its ability to decrease the level of intracellular ROS, lipid peroxidation, and NO levels in Caco-2 cells, where the oxidative stress was induced by H_2_O_2_. The antidiabetic activity was evaluated by measuring the ability of the hydrolysate to inhibit, in Caco-2 cells, the activity of the DPP-IV enzyme which is expressed on the cell membrane surface [23].

## 2. Materials and Methods

### 2.1. Chemicals and Sample Preparation

Tris-HCl, ethylendiaminetetraacetic acid (EDTA), Griess reagent, NaCl, 2,2-azino-bis-(3-ethylbenzothiazoline-6-sulfonic acid (ABTS), Trolox, Potassium Persulfate, azo 2,2′-azobis(2-methylpropionamidine) dihydrochloride (AAPH), Methanol, Fluorescein sodium, 2,4,6-Tris(2-pyridyl)-s-triazine (TPTZ), 1,1-Diphenyl-2-picrylhydrazyl radical (DPPH) and the Fluorometric Intracellular ROS Kit were from Sigma-Aldrich (St. Louis, MO, USA). The DPP-IV assay kit was from Cayman Chemicals (Ann Arbor, Michigan, USA), while Gly-Pro-amido-4-methylcoumarin hydrobromide (Gly-Pro-AMC) was from AnaSpec (Freemont, CA, USA). Lipid Peroxidation (MDA) Assay Kit was from Abcam (Cambridge, UK). Dulbecco’s modified Eagle medium (DMEM), fetal bovine serum (FBS), L-glutamine, phosphate buffered saline (PBS), penicillin/streptomycin, 24 and 96-well plates were from Euroclone (Milan, Italy). The fish protein hydrolysate (FPH) was produced following procedures already described by Kvangarsnes K. et al. [22]. Briefly, head-on, gutted rainbow trout fish were minced and hydrolyzed with Papain F6 and Bromelain 400 (0.05% *w*/*w*) at 50 °C for 1 h. After enzyme inactivation, the mixture was centrifuged to obtain the water-soluble protein phase representing the FPH. The hydrolysate was then freeze-dried, milled into powder and vacuum-packed (50 mbar) in low-density polyethylene bags.

### 2.2. FPH Ultrafiltration

Earlier in the analysis of biological activity, FPH was ultrafiltrated with 3 kDa cut-off Millipore UF System ultrafiltration (UF) membrane (Millipore, Bedford, MA, USA) using optimized conditions [24]. The recovered peptides solution was dried in a Speed-Vac (Martin Christ Gefriertrocknungsanlagen GmbH, Osterode am Harz, Germany) and stored at −80 °C until use.

### 2.3. Delineation of Potential Biological Activities

The open access BIOPEP-UWM database was employed to recognize potential bioactive peptides present in FPH (https://biochemia.uwm.edu.pl accessed on the 2 January 2023).

### 2.4. Cell Culture

Caco-2 cells were obtained from INSERM (Paris, France) and cultured in DMEM with 25 mM of glucose, 3.7 g/L of NaHCO_3_, 4 mM of stable L-glutamine, 1% nonessential amino acids, 100 U/L of penicillin, and 100 μg/L of streptomycin (complete medium), 10% heat-inactivated FBS at 37 °C in a 90% air/10% CO_2_ atmosphere.

### 2.5. MTT (3-(4,5-Dimethylthiazol-2-yl)-2,5-Diphenyltetrazolium Bromide) Assay

MTT assay was conducted as per the method of Lammi et al. [25]. In a 96-well, Caco-2 cells (3 × 10^4^) were seeded and treated with FPH (0.1 to 5 mg/mL) or vehicle (H_2_O) in complete DMEM for 48 h at 37 °C under a 5% CO_2_ atmosphere. Eventually, the treatment was eliminated and 100 µL/well of MTT (0.5 mg/mL) filtered solution was supplemented. After 2 h of incubation at 37 °C under a 5% CO_2_ atmosphere, the MTT solution was aspirated and 100 µL/well of the lysis buffer (8 mM HCl + 0.5% NP-40 in DMSO) was added. After 10 min of slow shaking, the Synergy H1 microplate reader was applied to read the absorbance at 575 nm (Biotek, Bad Friedrichshall, Germany).

### 2.6. TEAC (Trolox Equivalent Antioxidant Capacity) Assay

The TEAC assay, based on the reduction of the ABTS radical induced by antioxidants, was performed according to the method of Bollati et al. [26]. For the ABTS + ● preparations, 7 mM ABTS solution was mixed with 2.45 mM potassium persulfate (1:1) and stored for 16 h in the dark at room temperature (RT). The ABTS + ● was diluted in 5 mM phosphate buffer (pH 7.4) to obtain a stable absorbance of 0.700 (±0.02) at 730 nm in order to prepare the ABTS reagent. For the assay, 10 µL of FPH at the final concentrations of 0.1, 0.5 and 1.0 mg/mL were added to 140 µL of the diluted ABTS + ●. The microplate was incubated for 30 min at 30 °C, and the absorbance was read at 730 nm applying Synergy H1. A Trolox calibration curve (60–320 µM) was used to evaluate the TEAC values.

### 2.7. ORAC Assay

The ORAC assay, based on the scavenging of peroxyl radicals generated by AAPH, was performed following the Santos-Sánchez et al. method [27]. Briefly, 25 µL of FPH (at the final concentrations of 0.1, 0.5 and 1.0 mg/mL) was added to 50 µL sodium fluorescein (2.934 mg/L) and incubated for 15 min at 37 °C. Therefore, 25 µL of AAPH (60.84 mM) was added, and the fluorescence read at 485 nm ex. and 528 nm em. every 5 min for 120 min using a Synergy H1 microplate reader. The area under the curve (AUC) was calculated for each sample subtracting the AUC of the blank. The results were calculated using a Trolox calibration curve (2–50 µM).

### 2.8. Determination of Ferric Reducing Antioxidant Power

The FRAP assay evaluates the ability of a sample to reduce ferric ion (Fe^3+^) into ferrous ion (Fe^2+^). The FRAP test was carried out according to the method described by Bartolomei et al. [28] with a slight modification. Thus, 10 µL of FPH solutions (at the final concentrations of 0.1, 0.5, 1.0 and 2.5 mg/mL), were mixed with 140 µL of FRAP reagent (1.3 mL of a 10 mM TPTZ solution in 40 mM HCl, 1.3 mL of 20 mM FeCl_3_ × 6 H_2_O and 13 mL of 0.3 M acetate buffer (pH 3.6)). After 30 min incubation at 37 °C, the absorbance was read at 595 nm using a Synergy H1 microplate reader. The results were calculated by a Trolox standard curve obtained using different concentrations (3–400 µM).

### 2.9. Determination of DPPH Activity

The DPPH assay was performed to determine the antioxidant activity by standard method with a slight modification [29]. An aliquot (45 μL) of 0.0125 mM DPPH methanolic solution was added to 15 μL of FPH solutions (at final concentration of 0.1, 1.0 and 5.0 mg/mL). The mixture was incubated for 30 min in the dark at RT, and absorbance was measured at 520 nm using the Synergy H1 microplate reader.

### 2.10. Nitric Oxide Level Evaluation on Caco-2 Cells

The nitric oxide levels were assessed following the method of Bollati et al. [26]. Caco-2 cells (1.5 × 10^5^/well) were seeded on a 24-well plate and were treated the following day with FPH (final concentrations of 1.0 mg/mL) and incubated at 37 °C under a 5% CO_2_ atmosphere for 24 h. After incubation, cells were treated with H_2_O_2_ (1 mM) or vehicle (H_2_O) for 1 h; then, the cell culture media were taken and centrifuged at 13,000× *g* for 15 min to remove insoluble material. Nitric oxide determination was performed by the Griess reagent test. Briefly, 50 μL of the Griess reagent solution (1.0 g of Griess reagent in 25 mL of dH_2_O) were incubated with 50 μL of the culture supernatants for 15 min at RT in the dark. The absorbance was measured at 540 nm using the Synergy H1 microplate reader.

### 2.11. Measurement of Intracellular ROS

3 × 10^4^ Caco-2 cells/well were seeded on a black 96-well plate overnight in growth medium. The following day, the medium was discarded and replaced with 50 μL/well of fresh medium, 50 μL/well of master reaction mix (20 μL 500× ROS detection reagent stock solution + 10 mL assay buffer) were added, and the cells were incubated at 5% CO_2_, 37 °C for 1 h in the dark. Then, cells were treated for 24 h with 10 μL of 11x FPH to reach the final concentrations of 1 mg/mL at 37 °C in the dark. To induce ROS, cells were treated with H_2_O_2_ at a final concentration of 1 mM for 30 min at 37 °C in the dark, and fluorescence intensity (ex./em. 490/525 nm) was recorded using a Synergy H1 microplate reader.

### 2.12. Lipid Peroxidation (MDA) Assay

Lipid peroxidation lead to the degradation of lipids and can be detected by the reaction of MDA with thiobarbituric acid (TBA) to form a colorimetric product. Caco-2 cells (2.5 × 10^5^ cells/well) were seeded in a 24-well plate, and the following day, they were treated with 1 mg/mL of FPH for 24 h at 37 °C under a 5% CO_2_ atmosphere. The following day, cells were incubated with H_2_O_2_ 1 mM or vehicle (H_2_O) for 30 min. Cells were lysed with 150 µL ice-cold MDA lysis buffer (1.5 µL of butylated hydroxytoluene BHT 100×). Lysates were centrifuged at 13,000× *g* for 10 min; they were filtered through a 0.2 µm filter to remove insoluble material. To form the MDA-TBA adduct, 300 µL of the TBA solution was added into each vial containing samples, incubated at 95 °C for 60 min, and cooled to RT for 10 min in an ice bath. For analysis, 100 µL of each reaction mixture was pipetted into a 96-well plate, and the absorbance at 532 nm was measured by a Synergy H1 microplate reader.

### 2.13. In Vitro Measurement of the ACE Inhibitory Activity

ACE inhibitory activity was proven by measuring, with HPLC, the development of hippuric acid (HA) using hippuryl-histidyl-leucine (HHL) as a mimic substrate for ACE I. FPH was tested at 1 mg/mL in 100 mM Tris-HCOOH, 300 mM NaCl pH 8.3 buffer, and using ACE from porcine kidney (Sigma-Aldrich, Milan, Italy). All experimental details of sample preparations and analyses conditions have been published elsewhere [30,31].

### 2.14. In Vitro DPP-IV Activityinhibition Assay

The in vitro experiments were carried out in duplicate in a half volume 96-well solid plate (white) using conditions previously optimized [28]. Each reaction (50 µL) was prepared by adding the reagents in a microcentrifuge tube in the following order: 1 X assay buffer [20 mM Tris-HCl, pH 8.0, containing 100 mM NaCl, and 1 mM EDTA] (30 µL), FPH at final concentrations of 0.1, 0.5, 1 and 2.5 mg/mL or vehicle (10 µL), and, finally, the DPP-IV enzyme (10 µL). Afterward, added in the plate were 50 µL/well of samples and 50 µL/well of substrate solution (200 μM H-Gly-Pro-7-amido-4-methylcoumarin (AMC)) at 37 °C for 30 min. The fluorescence intensity was read using the Synergy H1 microplate reader (ex./em. 360/465 nm). The DPP-IV enzyme and the substrate solution were provided by Cayman Chemicals (Ann Arbor, Michigan, USA).

### 2.15. Evaluation of the Inhibitory Effect of FPH on Cellular DPP-IV Activity

A total of 5 × 10^4^ Caco-2 cells/well were seeded in black 96-well plates with clear bottoms. The second day after seeding, spent media was discarded, and the cells were treated with 0.5, 1, 2.5 and 5 mg/mL of FPH or vehicle (H_2_O) in growth medium for 3 h at 37 °C. The treatments were discarded; Caco-2 cells were washed with 100 μL of PBS (Ca^++^, Mg^++^ free); 100 μL of Gly-Pro-AMC substrate was added to each well of at the concentration of 20 μM in PBS (Ca^++^, Mg^++^ free). Fluorescence signal (ex./em. 350/450 nm) was recorded after 30 min using a Synergy H1 microplate reader. Experiments were carried out following previously optimized conditions [28].

### 2.16. Western Blot Analysis

Caco-2 cells (1.5 × 10^5^ well) were seeded on 24-well plates and incubated at 37 °C under 5% CO_2_ atmosphere. The next day, cells were treated with 1.0 mg/mL of FPH in complete growth medium for 24 h. The day after, cells were incubated with 1 mM H_2_O_2_ or vehicle (H_2_O) for 60 min. Caco-2 cell proteins were obtained using 40 µL ice-cold lysis buffer (RIPA buffer + inhibitor cocktail + 1:100 PMSF + 1:100 Na-orthovanadate + 1:1000 β-mercaptoethanol). Bradford’s method was applied to quantify the total protein content, and 50 μg of total proteins were loaded on a precast 7.5% SDS-PAGE gel at 130 V for 45 min. Subsequently, the gel was pre-equilibrated in H_2_O for 5 min at room temperature (RT) and transferred to a nitrocellulose membrane (Mini nitrocellulose Transfer Packs,) using a Trans-Blot Turbo at 1.3 A, 25 V for 7 min. Thereafter, the membranes were blocked with 5% milk or BSA and incubated overnight at 4 °C with primary antibodies anti-iNOS, anti-NRF2 and anti-β-actin. The next day, the membranes were washed and incubated with secondary antibodies conjugated with HRP for 1 h at RT and a chemiluminescent reagent was used to visualize target proteins. Their signal was quantified using the Image Lab Software (Biorad, Hercules, CA, USA). The internal control β-actin was used to normalize loading variations.

### 2.17. Statistical Investigation

All results were expressed as the mean ± standard deviation (SD), where *p*-values < 0.05 were considered to be significant. Statistical analyses were performed by one-way ANOVA, followed by Tukey’s post-test (Graphpad Prism 9, GraphPad Software, La Jolla, CA, USA).

## 3. Results

### 3.1. Bioactive Peptides Prediction by BIOPEP

In the previous work [22], the peptides present in the FPH hydrolysate were recognized. As reported in Table 1, using the open access database BIOPEP, we verified the presence in all 20 peptides of repeated amino acid sequences that refer to potential antioxidant, ACE inhibitory and DPP-IV activity.

### 3.2. Characterization of the Biochemical Activities of FPH

#### 3.2.1. Antioxidant Activity

The in vitro antioxidant activity of FPH was examined at 0.1, 0.5 and 1 mg/mL using the TAEC and ORAC assays, whereas the FRAP was examined in the range of 0.1–2.5 mg/mL, and the DPPH assay was verified in the range of 0.1–5.0 mg/mL. The FPH decreased the ABTS radical by 26.58 ± 3.04%, 60.96 ± 1.3% and 64.06 ± 1.3% at 0.1, 0.5 and 1.0 mg/mL, respectively (*p* < 0.0001, Figure 1A). Additionally, in the ORAC assay, this protein hydrolysate was capable of scavenging the peroxyl radicals produced by 2,2′-azobis (2-methylpropionamidine) dihydrochloride up to 81.1 ± 1.0%, 102.2 ± 1.13% and 103.4 ± 1.62, versus the control sample at 0.1, 0.5 and 1 mg/mL, respectively (*p* < 0.0001, Figure 1B,C), showing that the FPH increased the FRAP by 300 ± 20.41%, 981.3 ± 23.94%, 1625 ± 20.41% and 2625 ± 64.55% at 0.1, 0.5, 1 and 2.5 mg/mL, respectively. Finally, as shown by Figure 1D, the same hydrolysate reduced the DPPH radical by 24.65 ± 1.26%, 29.49 ± 1.73% and 50.72 ± 1.42 at 0.1, 1.0 and 5.0 mg/mL, respectively (*p* < 0.0001, Figure 1D). A dose-dependent response was observed in all the antioxidant assays.

#### 3.2.2. ACE and DPP-IV Inhibitory Activities

To gain a more thorough characterization of FPH multifunctional activities, its ACE inhibitory activity was examined. The inhibitory effects of FPH on the in vitro ACE activity were evaluated using the porcine recombinant form of the enzyme. Figure 2A shows that FPH inhibited the ACE activity by 55.6 ± 0.2% at 1035 µg/mL concentration. The graphic report of % ACE inhibition vs. Log_10_ FPH concentration calculation (Figure 2B) shows an IC_50_ value of 0.81 ± 0.0016 mg/mL. In agreement with our results, different studies have referenced the ACE-inhibitor activity of fish hydrolysates from species such as goby [32], salmon [33], lizard fish [34], Sardinella [35] and water fish [36].

Purified recombinant DPP-IV enzyme was used to evaluate, by preliminary biochemical experiments, the ability of FPH to modulate DPP-IV activity. The enzyme was incubated with the FPH in the concentration range of 0.1–2.5 mg/mL, and the fluorescent substrate, H-Gly-Pro-AMC, for 30 min at 37 °C. The reaction was supervised by measuring the fluorescence signals (465 nm) due to the release of the free AMC group after the cleavage of the peptide H-Gly-Pro, catalyzed by DPP-IV. Figure 2C shows that the FPH reduces DPP-IV activity in vitro by 1.58 ± 4%, 24.47 ± 2.75%, 40.69 ± 0.25% and 68.04 ± 3.6%, at 0.1, 0.5, 1 and 2.5 mg/mL, respectively, following a dose-response trend. Therefore, FPH reduces the DPP-IV activity, with IC_50_ of 1.34 ± 0.097 mg/mL.

### 3.3. Effect of FPH on the Caco-2 Cell Vitality

Cellular viability experiments (MTT assay) were conducted for excluding potential cytotoxic effects of the treatment with FPH on the Caco-2 cell line. After a 48 h treatment, no cytotoxic effect was observed up to 5 mg/mL versus control cells (C), indicating that FPH did not cause a cytotoxic effect in this dose range on Caco-2 cells (Figure 3).

### 3.4. Characterization of Biological Activities of FPH at Cellular Levels: Antioxidant and DPP-IV Inhibitory Activities

Oxidative stress, a condition that contributes to the onset of various pathological conditions, is given by a state of imbalance between endogenous oxidants and antioxidants. Notably, nitric oxide (NO) overproduction, in conjunction with other reactive oxygen species (ROS), contributes to oxidative stress. To evaluate the ability of FPH to modulate NO overproduction caused by H_2_O_2_, cellular experiments have been carried out. Our findings clearly demonstrate that Caco-2 cells treated with H_2_O_2_ alone showed an increase of NO levels up to 128.1 ± 6.64%. On the contrary, the presence of FPH brought intracellular NO values to baseline levels after a trigger of oxidative stress mediated by H_2_O_2_, resulting in a reduction of NO levels by 101.8 ± 11.81% (Figure 4A). Furthermore, for evaluating the protective activity of FPH on human intestinal Caco-2 cells, the fluorometric intracellular ROS assay has been performed. As shown in Figure 4B the H_2_O_2_ treatment results in the raise of intracellular ROS levels by 159.9 ± 9.25%, *versus* the control cells, which was attenuated by the pretreatment with FPH that decreases the ROS by 106.1 ± 6.58% at 1 mg/mL. These findings denote that the pretreatments with FPH preserve the Caco-2 cells against the increase of intracellular ROS induced by the H_2_O_2_ addition, thus restoring the ROS levels. These results are in agreement with a literate study in which fish hydrolysates scavenge oxygen free radicals, decrease ROS levels, and protect cells from free radical-induced cytotoxicity [37]. The process by which free radical species produce reactive intermediates is particularly evident in cell membranes and organelles, which contain different classes of lipids, as these are particularly susceptible to damage by ROS. These reactive intermediates can undergo further reactions such as malondialdehyde (MDA) and related compounds, known as TBA reactive substances (TBARS). MDA is a well-known biomarker in biological and medical sciences in the evaluation of oxidative stress correlated with various health problems [38,39]. Hence, the capacity of FPH to regulate the H_2_O_2_-induced lipid peroxidation in human intestinal Caco-2 cells was assessed by the MDA evaluation. According with the observed increase of NO and ROS after the H_2_O_2_ treatment, a remarkable increase of the lipid peroxidation was noted up to 144.6 ± 5.93% vs. the control cells. Moreover, our findings showed that the pretreatment with FPH (1 mg/mL) resulted in a reduction in MDA levels by 99.32 ± 16.75%, restoring the lipid peroxidation baseline levels (Figure 4C). These results are consistent with several investigations conducted on fish and fish byproduct hydrolysates regarding their antioxidant properties [40,41,42,43]. The inhibitory activity of FPH was then estimated in situ using Caco-2 cells, which express high levels of DPP-IV on their cellular membranes. Namely, Caco-2 cells were treated with FPH in the range of concentrations 0.5–5 mg/mL for 3 h, and the impacts on cellular DPP-IV activity were evaluated using the equal fluorescent substrate for 30 min at 37 °C. Figure 4D indicates that the FPH inhibited cellular DPP-IV activity by 5.12 ± 4.09%, 8.07 ± 1.25%, 16.61 ± 4.3% and 27.57 ± 3.7% at 0.5, 1, 2.5, and 5 mg/mL, respectively, validating the in vitro results.

### 3.5. FPH Modulate the iNOS and NRF2 Protein Levels in Caco-2 Cells

The effects of FPH on iNOS and NRF2 formation were evaluated on human intestinal Caco-2 cells after H_2_O_2-_induced oxidative stress by Western blot experiments. Results show that the H_2_O_2_ (1 mM) stimulation provoked a redox imbalance that led to a rise of the intracellular iNOS levels up to 123.6 ± 4.19% (Figure 5A). The pretreatment with FPH (1 mg/mL) reduced the H_2_O_2_-induced iNOS excess up to 103.5 ± 3.81% (Figure 5A).

In parallel, the effects of FPH on the NRF2 protein levels were gauged after oxidative stress initiation. Our results (Figure 5B) clearly demonstrated that, after H_2_O_2_ treatment (1 mM), the NRF2 protein decreased by 23.77 ± 4.24% in Caco-2 cells, whereas the pretreatment of Caco-2 cells with FPH (1 mg/mL) increased the H_2_O_2_-induced NRF2 protein up to 106.6 ± 5.09%, bringing its levels around the basal conditions (Figure 5B).

## 4. Discussion

Due to the broad bioactivity spectrum of marine peptides, the interest of the pharmaceutical and nutraceutical industry in their potential value for the treatment or prevention of various diseases has grown. In particular, it has been reported that marine bioactive peptides from several marine organisms act as antioxidants inhibiting lipid peroxidation, removing reactive oxygen species (ROS) and chelating pro-oxidant metal ions [44]. Although, in this research, the peptide fractions showed different antioxidant activity, most of the antioxidant peptides obtained from fish have a molecular weight between 0.5 and 1.5 kDa [45]. Under normal circumstances, there is a balance between ROS production and the activity of the antioxidant defense mechanisms. However, oxidative stress, which is an imbalance between pro-oxidants and antioxidants, acknowledged to be correlated with several metabolic pathologies. Lipids are extremely exposed to oxidation, considering that cell membranes are abundant in polyunsaturated fatty acids. One of the major causes of diabetic disorders triggered by hyperglycemia in patients with type II diabetes is increased oxidative stress related to a decline in cellular antioxidant defenses and lipid peroxidation in endothelial cells [46,47,48]. Moreover, the intensified ROS production can significantly promote peripheral insulin resistance, mitochondrial dysfunction, apoptosis and cell death, causing secretory dysfunction in pancreatic β-cells [49,50]. The condition of oxidative stress and diabetes is often also related with hypertension [51]. It has been reported that marine-derived peptides, in addition to having antioxidant activity, also exhibit ACE and dipeptidyl peptidase (DPP-IV) inhibitory properties together [52,53]. This could lead to their potential use as functional food ingredients for the management of hypertension and type II diabetes in a single therapy. Nowadays, ACE inhibitors are widely used as antihypertensive drugs, however, they have several negative side effects, such as allergic reactions, taste disturbances and skin rashes [54]; due to their high cost and limited physician experience, DPP-IV inhibitors are used on a limited basis [55]. Therefore, the research and application of natural ACE and DPP-IV inhibitor compounds with little or no side effects, and which can be consumed as part of the normal diet, are subjected to examination. In light of these observations, our study clearly demonstrated that FPH possesses multifunctional behavior being able to exert antioxidant, DPP-IV-, and ACE- inhibitory activities, respectively. This multitarget behavior may be explained considering the heterogeneous composition of the peptides that were identified within FPH sample [22]. Indeed, as elaborated in the Table 1, subjecting previously identified peptides to BIOPEP-WM (https://biochemia.uwm.edu.pl/biopep-uwm/ accessed on the 2 January 2023), many of them contain aminoacidic motives endowed with antioxidant, ACE, and DPP-IV inhibitory activities, respectively. Totally in agreement with these results, the biological investigation confirmed that FPH scavenges the ABTS and DPPH radicals, respectively, with an improvement of total ORAC and FRAP activities, respectively (Figure 1). These results are in line with the ability of FPH to reduce the H_2_O_2_-induced NO, ROS, and lipid peroxidation in human intestinal Caco-2 cells, respectively (Figure 3). These effects are clearly correlated with ability of peptide mixture to modulate the iNOS and NRF2 pathways, restoring their level towards physiological condition (Figure 5). In addition, our results clearly suggest that FPH drop the in vitro ACE activity with a dose-response trend and an IC_50_ equal to 0.81 ± 0.0016 mg/mL (Figure 2B). From a biochemical point of view, results indicated that FPH is less active in the inhibition of DPP-IV. Indeed, Figure 2C indicated that peptides reduce the enzyme activity, displaying an IC_50_ equal to 1.34 ± 0.097 mg/mL. In this context, DPP-IV is a relevant peptidase physiological expressed by many tissues, i.e., intestine [56,57]. Human intestinal Caco-2 cells are widely used for the study of peptides with DPP-IV inhibitory activity, as they represent a reliable and validated model [58,59,60]. In this study, findings clearly demonstrated that FPH maintain their ability to reduce the DPP-IV activity on Caco-2 cells, even with less activity. Comparable findings have been already obtained on plant-derived hydrolysates [58,59,61]. Protein hydrolysates showed better activity in cell-free than cell-based assays, even in these cases. This reduction in activity is probably due to the degradation that occurs by Caco-2 cells [58]. Indeed, many active proteases and peptidases that could metabolize dietary peptides by modulating their bioactivity are expressed in the intestinal brush border. Therefore, this organ not only allows the absorption of nutrients by acting as an important physiological barrier against the external environment, but also actively participates to the modulation of the physicochemical profiles of food protein hydrolysates, through the metabolic activity of its proteases. Overall, applying a multidisciplinary approach, this investigation offers in vitro evidence regarding the health-promoting activity of FPH, which could be the basis for the creation of new dietary supplements and/or functional foods. Of course, we are aware that this is the initial phase of this process and that, undoubtedly, an in vivo study on animal models is required to attain the proof of concept regarding the safety of the ingredients and health-promoting activity of this hydrolysate.

## Figures and Tables

**Figure 1 nutrients-15-00829-f001:**
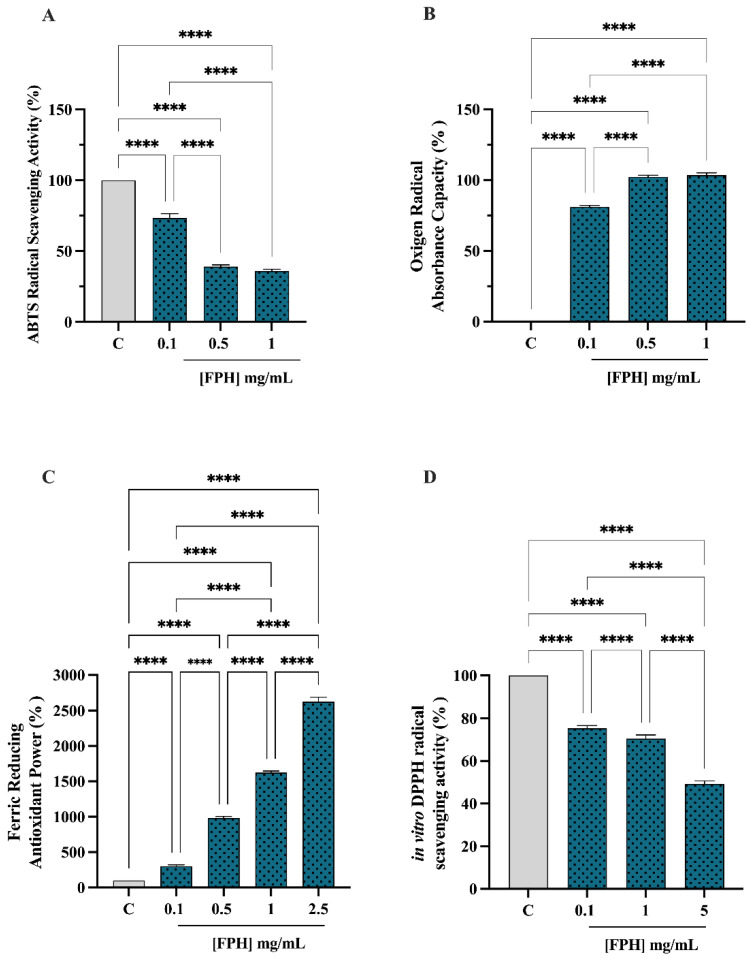
In vitro antioxidant power evaluation of the FPH by 2,2-azino-bis-(3-ethylbenzothiazoline-6-sulfonic acid (ABTS) (**A**), Oxygen Radical Absorbance Capacity (ORAC) (**B**), ferric reducing antioxidant power (FRAP) (**C**), and 2,2-diphenyl-1-picrylhydrazyl (DPPH) (**D**) assays. The data points represent the averages ± SD of three independent experiments performed in triplicate. All data sets were analyzed by one-way ANOVA followed by Tukey’s post-hoc test. C: control sample (H_2_O), ns: not significant, (****) *p* < 0.0001.

**Figure 2 nutrients-15-00829-f002:**
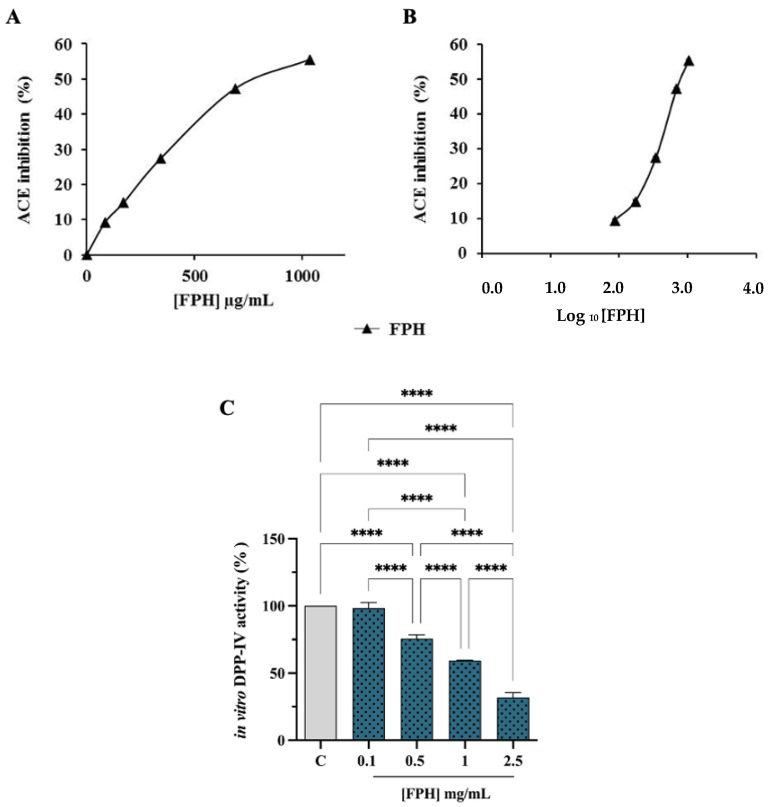
Evaluation of cell-free ACE and DPP-IV inhibitory activity of the FPH: results of % ACE inhibition vs. FPH concentration (µg/mL) (**A**); graphic report of % ACE inhibition vs Log_10_ FPH concentration; IC_50_ is the concentration needed to observe 50% inhibition of ACE activity (**B**); inhibition of the activity of human recombinant DPP-IV (**C**). The data are represented as the means ± SD of six independent experiments, performed in triplicate. Statistical analysis was performed by one-way ANOVA, followed by Tukey’s post-hoc test (****) *p* < 0.0001.

**Figure 3 nutrients-15-00829-f003:**
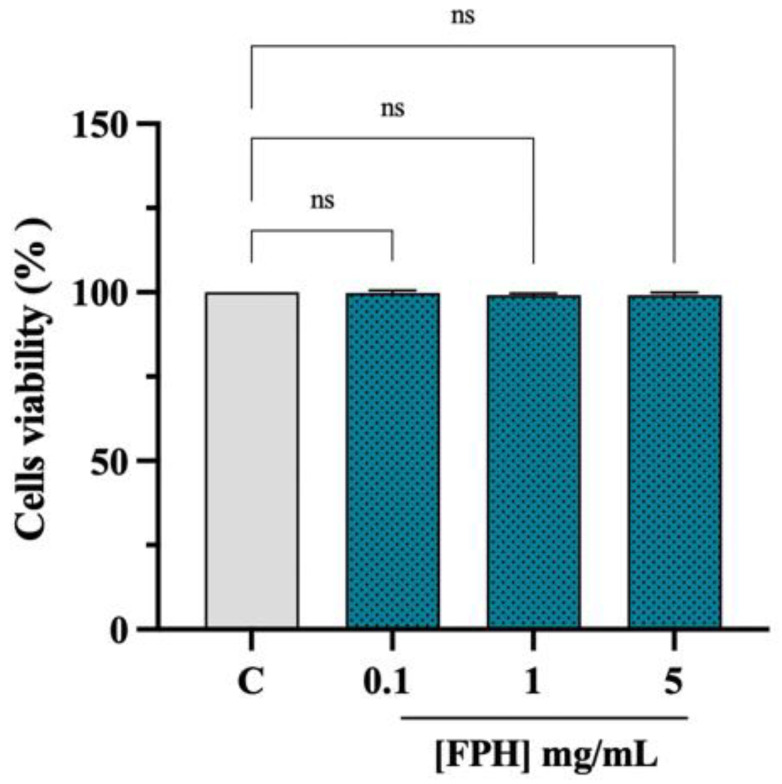
Caco-2 cell viability after FPH treatment. Bar graphs indicating the results of cell viability assay of Caco-2 cells after FPH (0.1–5 mg/mL) treatment for 48 h. The data points represent the averages ± SD of three experiments in triplicate, statistical analysis was performed by one-way ANOVA. C: control sample (H_2_O), ns: not significant.

**Figure 4 nutrients-15-00829-f004:**
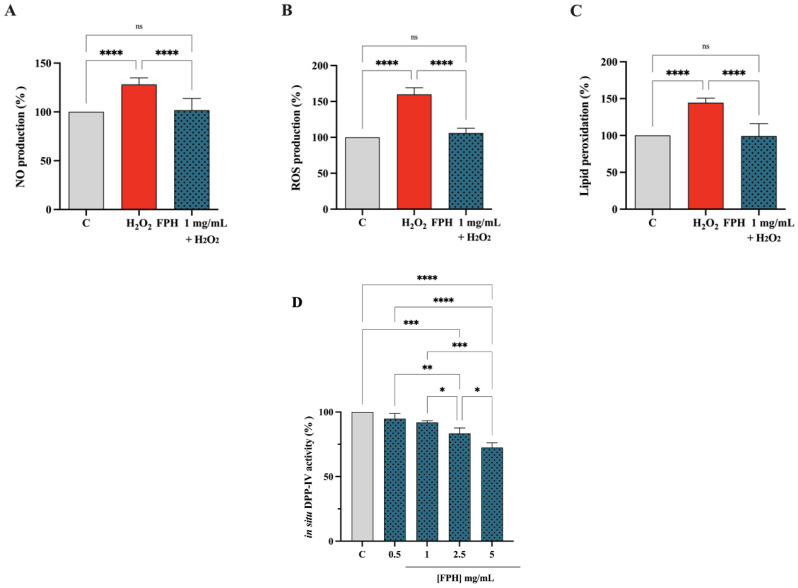
Effect of FPH on the H_2_O_2_-induced NO production in human intestinal cells (**A**). Modulation of intracellular ROS level H_2_O_2_-induced after the pretreatment with FPH (**B**). Modulation of lipid peroxidation H_2_O_2_-induced after the pretreatment with FPH (**C**). In situ inhibition of the DPP-IV activity expressed by nondifferentiated Caco-2 cells, after 3 h of treatment (**D**). The data are represented as the means ± SD of six independent experiments, performed in triplicate. Statistical analysis was performed by one-way ANOVA, followed by Tukey’s post-hoc test. ns: not significant; (*) *p* < 0.05, (**) *p* < 0.01, (***) *p* < 0.001, (****) *p* < 0.0001.

**Figure 5 nutrients-15-00829-f005:**
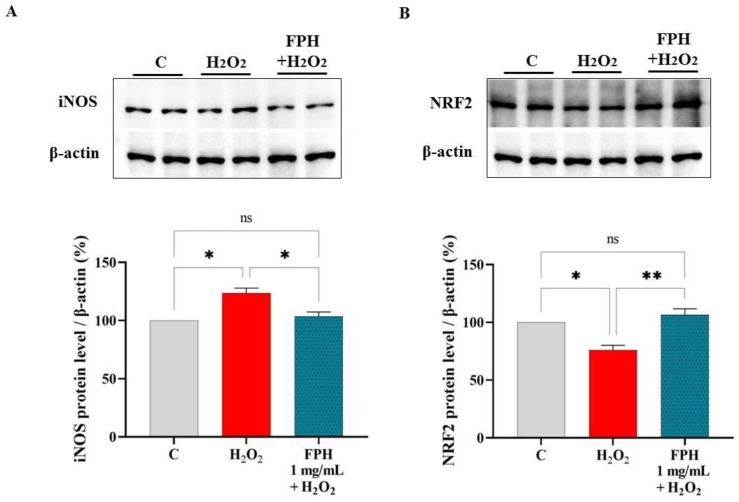
Effect of FPH on the H_2_O_2_ (1 mM)-induced iNOS (**A**) and NRF2 (**B**) protein levels in human intestinal Caco-2 cells. The data points represent the averages ± s.d. of three independent experiments in duplicate. All data sets were analyzed by one-way ANOVA followed by Tukey’s post-hoc test. C: control sample; ns: not significant; (*) *p* < 0.05, (**) *p* < 0.01.

**Table 1 nutrients-15-00829-t001:** Potential bioactivities according to BIOPEP database search.

Sequence	Potential Bioactive Peptides	Biological Functions
(K)TEPGSLPEGKVK(I).	VK, GS, GK, EG, PG, TE, GKV, LP LP, EP, SL, EG, KV, PG, TE, VK	ACE inhibitor DPP-IV inhibitor
(K)IRLESDGSLLDVDEDDVEK(A)	IR IR, GS, DG, VE, EK, SDGS LL, EK, SL, ES, IR, RL, VD, VE	Antioxidant ACE inhibitor DPP-IV inhibitor
(K)TELHFNHFAENSAFGIVPQPKSEDK(Q)	LH, EL, LHF AF, VP, GI, FG, TE, PQ, QP FA, VP, QP, AE, AF, FN, GI, HF, KS, LH, NH, PK, PQ, TE	Antioxidant ACE inhibitor DPP-IV inhibitor
(K)DLKRTKVLLADAQIMLDHMK(N)	LK LA, KR, DA LA, LL, AD, IM, KR, KV, MK, ML, QI, TK, VL	Antioxidant ACE inhibitor DPP-IV inhibitor
(K)QRPSSTTTDTGK(L)	RP, GK, TG, ST RP, PS, TD, TG, TT	ACE inhibitor DPP-IV inhibitor
(K)DCKKSRFSSDIVGPSDPQPDK(N)	RF, GP, VG, PQ, VGP, QP GP, QP, DP, KK, KS, PQ, PS, VG	ACE inhibitor DPP-IV inhibitor
(K)TPVESGASSAENRAADSTMTTSKPK(D)	KP RA, AA, GA, SG, KP, VE, TP, ST RA, TP, KP, GA, RA, AA, AD, AE, AS, ES, NR, PK, PV, SK, TM, TS, TT, VE	Antioxidant ACE inhibitor DPP-IV inhibitor
(K)LCAEPVAESAKSEHAVTEESETK(D)	TE, AV VA, HA, EP, AE, AV, EH, ES, ET, KS, PV, TE, TK, VT	ACE inhibitor DPP-IV inhibitor
(K)RGGITCFLKVKCEEEMINDTMK(L)	LK VK, GI, GG, CF, RG FL, GG, GI, IN, KV, MI, MK, ND, RG, TM, VK	Antioxidant ACE inhibitor DPP-IV inhibitor
(K)QHIIDGEKTIIQNPTDQQRKDHEK(A)	KD GE, DG, PT, EK NP, EK, DQ, GE, HE, HI, II, IQ, KT, PT, QH, QN, QQ, RK, TD, TI	Antioxidant ACE inhibitor DPP-IV inhibitor
(K)SEHEVQDAELRTLLQSSASRKTQK(K)	EL DA, QK, EV, LQ, TQ, AEL, LR LL, AE, AS, EH, EV, HE, KT, QD, QS, RK, TL, TQ, VQ	Antioxidant ACE inhibitor DPP-IV inhibitor
(K)IRCVEEKPVLSLPCVPHVAPPSNPK(A)	PHV, IR, KP IR, VAP, AP, VP, KP, VE, PP, EK, PH, LP PP, VA, AP, LP, VP, KP, NP, EK, SL, HV, IR, PH, PK, PS, PV, VE, VL	Antioxidant ACE inhibitor DPP-IV inhibitor
(K)EVIRLEKDPEMLK(A)	KD, IR, LK RL, IR, EV, EK, LEK EK, DP, EV, IR, ML, RL, VI	Antioxidant ACE inhibitor DPP-IV inhibitor
(K)ALYTQYLQFKENEIPLKETEK(S)	LY, LK LY, YL, PL, IP, EI, TE, LQ, TQ, EK, KE IP, EK, AL, PL, YT, EI, ET, KE, NE, QF, QY, TE, TQ, YL	Antioxidant ACE inhibitor DPP-IV inhibitor
(K)MSHKSAVANGGGPGNHAYLTNK(E)	AY PG, GP, YL, AY, GP, GG, NG, PG, NK, HK, AV GP, VA, HA, AV, AY, GG, KS, LT, NG, NH, PG, SH, TN, YL	Antioxidant ACE inhibitor DPP-IV inhibitor
(K)AISLNLNNYEK(E)	AI, NY, LN, EK, YE EK, SL, LN, NL, NN, NY, YE	ACE inhibitor DPP-IV inhibitor
(K)VSYECRVVSGKLVMGLDK(M)	GL, MG, GK, SG, SY, KL, YE, VM VV, GL, LV, MG, SY, VM, VS, YE, RV	ACE inhibitor DPP-IV inhibitor
(K)IVESYNTVSVLGVSK(S)	SVL GV, LG, SY, E, YN, LGV ES, GV, NT, SK, SV, SY, TV, VE, VL, VS, YN	Antioxidant ACE inhibitor DPP-IV inhibitor
(K)GSLGPFGVPGQVGPK(G)	LGP, GP, VP, VG, FG, GS, GV, GQ, LG, PG, VPG, QVGP GP, VP, SL, GV, PF, PG, PK, QV, VG, GPF	ACE inhibitor DPP-IV inhibitor
(K)GLQGSPGPMGKEGDVGPLGDAGGPGSKGEK(G)	GPL, PLG, GP, PL, VG, GL, AG, KG, DA, GS, MG, GK, GE, GG, QG, LG, GD, EG, PG, LQ, EK, KE, GPM, VGP, GP GP, SP, EK, GL, PL, AG, EG, GE, GG, KE, KG, MG, PG, PM, QG, SK, VG, GPM	ACE inhibitor DPP-IV inhibitor

## Data Availability

Not applicable.

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
