# Peer review of "Rainbow Trout (Oncorhynchus mykiss) as Source of Multifunctional Peptides with Antioxidant, ACE and DPP-IV Inhibitory Activities"

_nutrients, 2023, doi:10.3390/nu15040829_

Round 1

Reviewer 1 Report

An interesting experimental article based on an old & new hypothesis of metabolic effect of the small peptides from marine sources. 

The methodology is very well presented, the results are clear and discussion and conclusion are in line with results.

Over all a good paper, a fundamental research which can be part of future development regarding healthy life, alimentation and new opportunities for treatment  

Reviewer 2 Report

This paper is the most basic study of active peptides. The author should conduct in-depth study on the structure-activity relationship of a certain activity. From the separation and purification of polypeptides, the composition and location of amino acids, as well as the mechanism of action and signal pathway at the cellular level. The innovation of the paper needs to be greatly improved

Reviewer 3 Report

The authors investigated the antioxidant, ACE, and DPP-IV inhibitory activities of rainbow trout protein hydrolysate (FSH) in this study (Oncorhynchus mykiss). Using in vitro assays, the authors determined the biochemical properties of FPH. The authors discovered that FPH increased ACE and DPP-IV inhibitory activities while decreasing ROS and lipid peroxidation. The manuscript was well written, and the experiments were appropriate.

The manuscript should address the following questions: 

1.  Paper title is broad and does not adequately describe the paper's purpose. In the present study, the antioxidant properties of Rainbow trout fish protein hydrolysate were examined. The title should reflect the aim of the paper and be modified accordingly.

2. A reference number 13 (Kvangsrsnes K et al.) was given for FPH production procedures, which have not yet been published. As an additional reference, it is better to provide an appropriate published protocol.

3. It is not clear whether the authors used whole hydrolase or any fraction of hydrolase used in the present study. Elaborate the process in the methods section.

4. In the current study, the authors did not review literature or cite relevant papers. The preparation of FPH from Rainbow trout fish and its evaluation of antioxidant properties are available in the literature. For instance https://doi.org/10.1016/j.lwt.2019.03.049, https://doi.org/10.1111/ijfs.13587

5. To strengthen the paper, the authors validate the antioxidant properties of FPH in at least one more cell line, like immune cells (THP-1), which are commonly involved in inflammation in diseases such as metabolic and cardiovascular diseases.

6. Data on Caco-2 cell viability for the FPH concentrations used in the paper should be provided.

7. The majority of the methods lack relevant references (such as MTT, TEAC, ORAC, etc.) and proper information (L135 FPH pretreatment time, L145 composition of "Master reaction mixture," etc.). The authors update the methods with references.

8. L197 the title is inappropriate and should be changed in accordance with the text.

9. L205 section 3.2.1 the paragraph contain typos and incorrectly labeled figure numbers. Like L209, L213. Modify the section with proper figure numbers.

10. In the manuscript, authors must thoroughly check for typos and spacing errors. For example L288, L302

Round 2

Reviewer 2 Report

OK

Reviewer 3 Report

The authors improved the manuscript by including western blot data and proper citations in the text. The author's explanations for the questions were adequate. More experiments are needed to investigate the therapeutic applications of FPH, which was mentioned as a paper limitation.